# Psoriasis and Liver Damage in HIV-Infected Patients

**DOI:** 10.3390/cells10051099

**Published:** 2021-05-04

**Authors:** Carmen Busca Arenzana, Lucía Quintana Castanedo, Clara Chiloeches Fernández, Daniel Nieto Rodríguez, Pedro Herranz Pinto, Ana Belén Delgado Hierro, Antonio Olveira Martín, María Luisa Montes Ramírez

**Affiliations:** 1HIV Unit. Department of Internal Medicine, La Paz University Hospital, 28046 Madrid, Spain; anitadelgadohierro@hotmail.com (A.B.D.H.); mluisa.montes@salud.madrid.org (M.L.M.R.); 2Department of Dermatology, La Paz University Hospital, 28046 Madrid, Spain; lucia.quintana@salud.madrid.org (L.Q.C.); clara.chiloeches@salud.madrid.org (C.C.F.); daniel.nieto@salud.madrid.org (D.N.R.); pedro.herranz@salud.madrid.org (P.H.P.); 3Hepatology Unit, Department of Gastroenterology, La Paz University Hospital, 28046 Madrid, Spain; antonio.olveira@salud.madrid.org

**Keywords:** psoriasis, HIV, NAFLD, liver fibrosis

## Abstract

*Background/objectives*: Psoriasis is the most frequent skin disease in HIV-infected patients. Nonalcohol fatty liver disease (NAFLD) is more prevalent in patients with psoriasis. We report the prevalence of psoriasis and NAFLD and investigate risk factors of liver damage in HIV-infected patients with psoriasis. *Methods*: We performed a retrospective observational study. Steatosis was defined as indicative abdominal ultrasound findings, CAP (controlled attenuated parameter by transient elastography) > 238 dB/m, and/or triglyceride and glucose index (TyG) > 8.38. Significant (fibrosis ≥ 2) and advanced liver fibrosis (fibrosis ≤ F3) were studied by transient elastography (TE) and/or FIB-4 using standard cutoff points. FIB-4 (Fibrosis 4 score) results were adjusted for hepatitis C (HCV)-coinfected patients. *Results*: We identified 80 patients with psoriasis (prevalence, 1.5%; 95% CI, 1.1–1.8). Psoriasis was severe (PASI > 10 and/or psoriatic arthritis) in 27.5% of cases. The prevalence of steatosis was 72.5% (95% CI, 65–85). Severe psoriasis was an independent risk factor for steatosis (OR, 12; 95% CI, 1.2–120; *p* = 0.03). Significant liver fibrosis (*p* < 0.05) was associated with HCV coinfection (OR 3.4; 95% CI, 1.1–10.6), total CD4 (OR 0.99; 95% CI, 0.99–1), and time of efavirenz exposure (OR 1.2; 95% CI, 1.0–1.3). *Conclusions*: The prevalence of psoriasis in HIV-infected patients was similar to that of the general population. Steatosis is highly prevalent, and severe psoriasis is an independent risk factor for steatosis in HIV-infected patients.

## 1. Introduction

Chronic human immunodeficiency virus (HIV) infection is increasingly prevalent owing the advent of antiretroviral therapy (ART), which has reduced mortality and the number of AIDS-defining diseases, thus considerably improving survival. However, non-HIV-associated comorbid conditions, such as cardiovascular events and non-AIDS-defining malignant tumors, are becoming increasingly frequent [1]. This phenomenon is not only due to the increase in survival; HIV-infected persons maintain marked long-term immunological activation, despite the fact that their disease is controlled with ART [2].

Psoriasis is a chronic inflammatory disease that affects 1–4% of the world’s population (estimated at 2.3% in Spain) [2]. It can be a form of presentation of HIV infection, its symptoms are very varied and atypical, it is usually severe, and its prevalence is similar to that recorded in the general population [3]. Psoriasis can appear at any stage of HIV infection, and several variations of the disease can occur simultaneously, a finding that is somewhat characteristic of this population. In fact, it was recently reported that HIV infection is an independent risk factor for developing psoriasis [4].

Psoriatic arthritis, which affects one-third of patients with seronegative psoriasis, is usually much more prevalent, severe, and refractory in HIV-infected persons. Several hypotheses have been put forward with respect to the association between HIV infection and psoriasis, the main one being that of a common potential genetic substrate and pathophysiology. Dysregulation in T lymphocytes (mainly as a result of systemic depletion of regulatory T cells and in other situations through viral RNA replication within the CD4^+^ dermal dendritic cell) could play an important role since the decrease in CD4 lymphocytes would activate a series of harmful proinflammatory pathways in the development of psoriasis. Furthermore, it is believed that HIV infection could play a direct role in psoriasis by acting as a co-stimulator of antigen presentation, which explains the presence of severe forms of psoriasis in patients with uncontrolled HIV infection [3].

Patients with psoriasis more frequently have cardiovascular risk factors, such as obesity, dyslipidemia, and a greater risk of developing metabolic syndrome, all of which are associated with a higher frequency of cardiovascular events and, therefore, increased mortality. Several studies have highlighted that NAFLD is more prevalent in patients with psoriasis than in the general population and that psoriasis is an independent risk factor for the development of NAFLD, independently of metabolic syndrome and obesity [5] Currently, NAFLD is considered the hepatic manifestation of metabolic syndrome, since it is the most common cause of chronic liver disease in the general population [6]. Thus, liver involvement in metabolic syndrome has been widely assessed in patients with psoriasis, although few studies have examined this involvement in HIV-infected patients, in whom pathophysiology is more complex. A series of factors associated with HIV infection contribute to the development of fatty liver, including lipodystrophy, ART, and mitochondrial toxicity. Despite the fact that psoriasis is the most common skin disease in HIV-infected patients and that NAFLD is currently the most common cause of liver disease, there are very few data on the association among these three entities.

The objective of the present study was to report the prevalence of psoriasis and NAFLD and to analyze which factors are associated with the liver disease in HIV-infected patients with psoriasis in the era of highly active antiretroviral therapy (HAART).

## 2. Materials and Methods

We performed a retrospective observational study at a single tertiary hospital. We reviewed the clinical histories of HIV-infected patients followed at the center and selected those that included a diagnosis of psoriasis at any point during follow-up.

We collected all data related to HIV infection, hepatitis C virus (HCV) coinfection, comorbidity, ART, and other medication, (approved. protocol codeHIV-La Paz cohort; HULP code: PI-890) Exposure time to non-nucleoside reverse transcriptase inhibitors (efavirenz and rilpivirine) and raltegravir was specifically analyzed because of their potential role in steatosis or liver damage. The dermatology history of each patient was reviewed by two dermatologists at our center. The type of psoriasis, the presence of psoriatic arthritis, and the severity thereof were reviewed, and a maximum Psoriasis Area Severity Index (PASI) was assigned to each case. Psoriasis was considered to be severe if the maximum PASI was >10 and/or the patient had psoriatic arthritis. 

Associated liver disease was studied by collecting all the results of abdominal ultrasound, transient elastography (TE) with controlled attenuated parameter (CAP), and liver biopsy, where available. The most recent laboratory findings in the clinical history were also reviewed in order to calculate noninvasive indices of steatosis (hepatic steatosis index (HSI) and the triglyceride/glucose index (TyG)) and liver fibrosis (AST-to-platelet ratio index (APRI) and Fibrosis 4 score (FIB-4)). Steatosis was defined as indicative abdominal ultrasound findings, CAP > 238 dB/m, and/or TyG > 8.38 [7,8]. Fibrosis was considered significant (F ≥ 2) if the transient elastography finding was ≥7.1 kPa and/or FIB-4 was >1.3 or 1.45 for patients without or with HCV coinfection, respectively; advanced liver fibrosis (F ≥ 3) was defined as TE ≤ 9.6 kPa and/or FIB-4 ≥ 2.67 or ≥3.25 for patients without or with HCV coinfection, respectively [9,10].

We performed a descriptive analysis of all the variables, both for the sample as a whole and for the subgroups (mild–moderate and severe psoriasis). Variables were summarized as proportions for categorical variables and as medians and 25–75th percentiles (IQR) for continuous variables. The comparisons between categorical variables were made by the Fisher test, and the comparisons between continuous variables were made by Mann–Whitney test. A multivariable logistic regression analysis was performed to establish associated factors of steatosis, significant fibrosis, and advanced fibrosis. The entermultivariable analysis was adjusted for age, time since diagnosis HIV infection, coinfection with HCV, metabolic syndrome, exposure to efavirenz (EFV) (years), and total CD4 (variables with a *p*-value < 0.1 in univariate analysis and/or clinically relevant) to control for residual confounding between variables. Furthermore, the stepwise multivariable method was performed to determine factors associated with steatosis, significant fibrosis, and advanced fibrosis.

This study was reviewed and approved by Ethical Committee for Clinical Research of the La Paz University Hospital (C.E.I.C) with research project title “Retrospective longitudinal descriptive registry of naive patients included in the cohort of patients with HIV infection at Hospital La Paz (Madrid)”; HIV-La Paz cohort; HULP code: PI-890.

## 3. Results

### 3.1. Study Population and Main Characteristics

We identified a total of 80 patients diagnosed with psoriasis from a single-center Spanish cohort of 5452 chronically HIV-infected patients, i.e., a prevalence of 1.5% (95% CI, 1.1–1.8%). Table 1 summarizes the main characteristics. Most patients were white men with a median age of 50 years and good immunological–virological control and became infected with HIV through homosexual relations.

All subjects were receiving antiretroviral therapy, integrate-inhibitor-based regimens were the most prevalent (37%). There were no differences in previous exposure to thymidine-NRTI, protease inhibitors, and efavirenz between mild–moderate and severe groups. Analysis of exposure time to rilpivirine and raltegravir showed no differences between the groups. However, in those who were exposed to EFV at any time, steatosis was associated with time of EFV exposure (1.1 vs. 6.7 years, *p* < 0.05). We found a significant association between the time of efavirenz exposure and estimated liver fibrosis by FIB-4 or TE for significant liver fibrosis (time of EFV exposure 3.7 vs. 9 years, *p* < 0.05) and a tendency for advanced liver fibrosis (time of EFV exposure 4.5 vs. 7.4 years, *p* = 0.43).

A total of 29 patients (36.7%) had HCV coinfection; of these, 22 were treated (sustained viral response in 21 (95.5%)), and the remaining seven were lost to follow-up or died before receiving treatment for HCV infection. Eleven patients were exposed to interferon-based therapy (45.8%); of these, three were diagnosed with psoriasis before treatment with interferon. Fourteen patients had cirrhosis, which was more prevalent in the subgroup of patients coinfected with HCV than in non-coinfected patients (12 (41.4%) vs. 2 (3.9%); *p* < 0.001).

### 3.2. Severity of Psoriasis

Plaque psoriasis was the most frequent type (74%), followed by guttate psoriasis (2.5%) and seborrheic psoriasis (2.5%). Severe psoriasis was recorded in 22 patients (27.5%), of whom 14 (17.7%) had a PASI > 10 and 10 (12.5%) had joint involvement. Moreover, 22% of patients with joint involvement had a PASI > 10. The median length of follow-up from diagnosis of HIV infection was 26 years in patients with severe psoriasis compared with 16 years in those with mild–moderate psoriasis (*p* < 0.05). Significant differences between severe and mild–moderate psoriasis were found for median glucose (101.5 mg/dL vs. 92 mg/dL), triglycerides (135 mg/dL vs. 103 mg/dL), the percentage of patients with metabolic syndrome (45% vs. 21%), and nadir CD4^+^ lymphocyte count (124.5 cells/mm^3^ vs. 230 cells/mm^3^) (Table 1). There were no differences with respect to HCV coinfection or in the use of lipid-lowering agents or hypoglycemic agents between the groups.

Subjects with severe psoriasis received systemic therapy in 73% (acitretin 54%, methotrexate 23%, etanercept 18%, and anti-TNF 18%). Only one subject received methotrexate in the mild–moderate psoriasis group, while five received it in the severe group, thereby precluding further analysis.

### 3.3. Liver Disease and Factors Associated with Liver Damage

Abdominal ultrasound data were available for 43 patients, and steatosis was diagnosed in 21% of patients. With respect to noninvasive serology indices, we observed a triglyceride/glucose index ≥ 8.38 in 62% and hepatic steatosis index ≥ 36 in 33.8% (38% of patients had HIS ≥ 36 in the severe subgroup vs. 32% in the mild–moderate subgroup (*p* = 0.87) and 91% had TyG ≥ 8.38 in the severe subgroup vs. 51% in the mild–moderate subgroup (*p* < 0.05)). The median (IQR) CAP value in the 28 patients with available measures was 235 dB/m (220–281), and 46.4% had a value ≥ 238 dB/m.

Irrespective of the diagnostic method used, the prevalence of steatosis in HIV-infected patients with psoriasis was 72.5% (95% CI, 62–81.4%). A total of 95.5% patients with severe psoriasis had steatosis, compared with 64% in the mild–moderate subgroup (*p* = 0.004).

TE data were available for 47 patients. Of these, 47% had a median value ≥ 7.1 kPa and 36% had a median value ≤ 9.6 kPa. The prevalence of significant liver fibrosis, irrespective of the diagnostic method used, was 43% (95% CI, 32.7–54.6%): 64% in patients with severe psoriasis and 35% in those with mild–moderate disease (*p* < 0.05). Irrespective of the diagnostic method used, the prevalence of advanced liver fibrosis was 22% (95% CI, 14.1–32.7%) (severe 41% vs. non-severe 15%; *p* < 0.05). Of the 11 available liver biopsies, steatosis was observed in 80% and advanced fibrosis in 36%.

The multivariable analysis of steatosis revealed severe psoriasis to be an independent risk factor for liver steatosis, with an OR adjusted for age, metabolic syndrome, HCV coinfection, years since diagnosis of HIV, total CD4 cell count, and time of efavirenz exposure of 12 years (95% CI, 1.2–120; *p* = 0.03). (Table 2).

The independent risk factors for significant liver fibrosis (≥F2) were HCV coinfection and time of exposure to efavirenz; CD4 cell count was a protective factor. The multivariate model of advanced liver fibrosis showed HCV coinfection as the main risk factor (OR, 20; 95% CI, 4–102; *p* < 0.001); CD4 cell count was a protective factor (Table 2).

## 4. Discussion

This was a retrospective observational study to analyze the prevalence, severity, and associated factors of psoriasis and NAFLD in a large cohort of immunologically and virologically controlled HIV-infected patients. We found the prevalence of psoriasis in HIV-infected patients to be similar to that of the general population, despite recent reports that HIV infection could be a risk factor for psoriasis [4]. Both severe psoriasis and psoriatic arthritis were found in a significant percentage of patients (27% and 12.5%, respectively).

Irrespective of the diagnostic method used, the prevalence of steatosis in our cohort was 72.5% (95% CI, 62–81.4%). This result is much higher than the prevalence of NAFLD in the general population with psoriasis, which is 43% according to data from a cohort of patients aged > 55 years [11]. Our results are also noteworthy if we compare them with the estimated prevalence of NAFLD in HIV-monoinfected patients without psoriasis, which is 35% [1]. All patients with severe psoriasis had steatosis, and 45% of them fulfilled the criteria for metabolic syndrome. In fact, severe psoriasis and metabolic syndrome were independent associated factors of steatosis. This finding agrees with data from the meta-analysis by Candia et al., which included more than 50,000 patients and reported a higher risk of NAFLD in patients with severe psoriasis than in those with less severe forms, albeit with an OR of 2.07 (95% CI, 1.59–2.71), which is much lower than the value we report in the present study, probably due to the smaller number of subjects included in our cohort [12]. 

It is important to note the association of time of exposure to efavirenz with liver steatosis as independent factor; efavirenz is a non-nucleoside retrotransciptase inhibitor against HIV that has been implied in different models of liver damage, including liver steatosis [13,14]. In other words, we observed not only that HIV-infected patients with severe psoriasis have a greater risk of NAFLD, but also that having chronic HIV infection and exposure to efavirenz implies a greater risk if we compare it with not being infected.

Of note, a few of the patients in our cohort were obese, and the median BMI was 25, without differences regarding psoriasis severity. This finding contrasts with that of a recent meta-analysis, which showed a direct correlation between obesity and severity of psoriasis and emphasized that patients with severe psoriasis are more obese than those with milder forms (OR 2.23 vs. 1.46) [12,15]. In our study, metabolic syndrome was present in 27% of patients—similar to the prevalence found in the non-psoriatic general population—and only in patients with severe psoriasis was the prevalence of metabolic syndrome as high as that reported in patients with psoriasis [16]. HIV-infected subjects with metabolic syndrome appear to have a different profile, with alterations in lipids, blood glucose, and hypertension being the most frequently observed criteria.

Using noninvasive serological markers and/or TE for diagnosis, we found the prevalence of liver fibrosis to be high; it was both significant and advanced. Liu et al. applied noninvasive methods for the diagnosis of fibrosis in HIV-infected patients with NAFLD and found that 4.3% had significant fibrosis [17]. In our cohort, this prevalence was much higher and was even significantly higher in the subgroup of patients with severe psoriasis than in those with non-severe psoriasis. This important difference is justified by the high prevalence of HCV coinfection. In fact, HCV coinfection was the main risk factor for liver fibrosis in both multivariable fibrosis models. Furthermore, time of efavirenz exposure was independently associated with significant fibrosis. Both factors contributed in an additive manner to the liver fibrosis in our cohort. We found no association with metabolic factors, in contrast with the general population, where a previous history of metabolic syndrome was associated with the presence of liver fibrosis [15].

The main limitations of the present study are its retrospective design, the absence of non-serological tests in all patients (e.g., abdominal ultrasound and CAP), and the absence of liver biopsy specimens for the diagnosis of NAFLD and liver fibrosis, together with the high percentage of patients with hepatitis HCV, which significantly modifies liver damage. In addition, alcohol intake was not able to systematically quantified from retrospective clinical records. Another significant limitation was the lack of a control group featuring HIV-uninfected subjects with psoriasis. Nevertheless, given that the frequency of hepatitis C coinfection was similar in patients with severe and mild-moderate psoriasis and the adjustments of the multivariable analysis were robust, we believe that our results are both reliable and relevant.

## 5. Conclusions

In conclusion, the prevalence of psoriasis in our cohort of patients with well-controlled chronic HIV infection was similar to that found in the general population. However, the presence of steatosis was much more prevalent, even without the co-occurrence of metabolic syndrome or obesity; this could not be completely explained by poorer chronic inflammation, as during the period prior to the availability of combined antiretroviral treatment. Severe psoriasis was a determinant of significant liver steatosis, irrespective of the presence of HCV coinfection. Our results highlight the need for more exhaustive control of risk factors associated with NAFLD, especially in HIV-infected patients with severe psoriasis.

## Figures and Tables

**Table 1 cells-10-01099-t001:** Clinical characteristics.

Variables		All	Mild–Moderate Psoriasis	Severe Psoriasis	*p*-Value
		*N* = 80	*N* = 68	*N* = 22	
Age (years) *		50 (43–55)	45 (43–56)	52 (46–55)	NS
Female gender, *N* (%)		17 (21)	14 (24)	3 (14)	NS
Origin (Spain), *N* (%)		73 (92)	53 (93)	20 (91)	NS
C AIDS stage, *N* (%)		9 (11)	5 (9)	4 (18)	NS
HCV-coinfection, *N* (%)		29 (37)	18 (31)	11 (50)	NS
Transmission route, *N* (%)	Men having sex with men	29 (38)	25 (44)	4 (20)	0.02
	Heterosexual	16 (21)	14 (25)	2 (10)	
	UDVP	27 (35)	16 (28)	11 (55)	
	Transfusion	1 (1)	1 (2)	0 (0)	
	Hemophile	2 (3)	0 (0)	2 (10)	
	Unknown	2(3)	1 (2)	1 (5)	
HIV infection controlled, *N* (%)		79 (99)	58 (100)	21 (94)	NS
HIV infection time (years) *		19 (8–25)	16 (8–23)	26 (13–30)	0.01
CD4 cell count (cells/µL) *		676 (481–898)	746 (581–920)	520 (282–670)	<0.01
Nadir CD4 cell count (cells/µL) *		209 (111–293)	230 (160–325)	124 (70–194)	0.01
Type of current cART, *n* (%)					
PI-based		26 (32)	19 (33)	7 (32)	
NNRTI-based		24 (30)	16 (28)	8 (36)	NS
INI-based		30 (37)	23 (40)	7 (32)	
BMI (kg/m^2^) *		25 (22–27)	24 (22–27)	26 (23–27)	NS
Metabolic syndrome, *N* (%) ^&^		22 (27)	12 (21)	10 (45)	0.05
ALT (UI/L) *		27 (19–40)	27 (18–38)	27 (21–46)	0.5
AST (UI/L) *		25 (21–31)	25 (22–31)	26 (21–35)	NS
GGT (UI/L) *		30 (19–53)	26 (18–51)	34 (28–63)	0.04
FA (UI/L) *		81(64–102)	72 (62–94)	88 (77–120)	0.02
Bilirubin (mg/dL) *		0.6 (0.4–0.8)	0.5 (0.4–0.8)	0.7 (0.5–0.8)	NS
Glucose (mg/dL) *		94 (86–105)	92 (85–101)	101 (91–117)	0.02
Cholesterol (mg/dL) *		180 (153–206)	180 (153–207)	179 (154–201)	NS
Triglycerides (mg/dL) *		117 (83–165)	103 (77–160)	135 (115–179)	0.02
Diabetes mellitus or abnormal fasting glucose, *N* (%)		12 (15.4)	6 (10.7)	6 (27.3)	0.09
Other drugs	Lipid-lowering drugs	21 (27.3)	13 (23.6)	8 (36.4)	NS
	Glucose-lowering drugs	11 (14.1)	5 (8.9)	6 (27.3)	0.06
**Liver Steatosis**, *N* (%)		59 (72.5)	37 (64)	21 (95.5)	<0.01
Ultrasound, *N* (%)		9 (21)	5 (19)	4 (24)	NS
Noninvasive markers	HSI *	34 (30–38)	33 (30–37)	35 (30–40)	NS
	TyG *	8.7 (8.3–9.1)	8.5 (8.2–8.9)	8.9 (8.5–9.4)	<0.01
	CAP *	235 (220–284)	235 (221–279)	238 (220–286)	NS
**Liver Fibrosis**					
Transient elastography (kPa) *		6.3 (4.8–14.3)	5.7 (4.4–10.2)	9.8 (6.1–14.6)	0.07
Noninvasive markers FIB-4 ≥ 2.67, *N* (%)		9 (11.8)	4 (7.4)	5 (22.7)	NS
APRI > 1.5, *N* (%)		6 (7.7)	3 (5.4)	3 (13.6)	NS
Significant fibrosis, *N* (%) ^Ç^		33 (43.4)	19 (35.2)	14 (63.6)	0.04
Advanced liver fibrosis, *N* (%) ^$^		17 (22.4)	8 (14.8)	9 (40.9)	0.03

* Medians (25–75th percentiles). ^&^ Metabolic syndrome: diagnosis criteria by Adult Panel Treatment (ATP) III definition: three of the five risk abnormalities (waist circumference ≥ 102 cm in men or ≥88 cm in women, fasting glucose ≥ 100 mg/dL or on treatment, triglycerides ≥ 150 mg/dL, HDL < 40 mg/dL in men and <50 mg/dL in women or on treatment, and systolic blood pressure ≥ 130 mmHg and/or diastolic blood pressure ≥ 85 mmHg or on treatment). ^Ç^ Liver fibrosis measured by transient elastography ≥7.1 kPa and/or FIB-4 > 1.3 for patients without HCV infection and 1.45 for patients with hepatitis C coinfection. ^$^ Advanced fibrosis measured by ET ≥ 9.6 kPa and/or FIB-4 ≥ 2.67 for patients without HCV infection and ≥3.25 for patients with hepatitis C coinfection.

**Table 2 cells-10-01099-t002:** Multivariable logistic regression.

Variables	Univariable Analysis		Enter Model		By Step Model	
	OR (95% CI)	*p*	OR (95% CI)	*p*	OR (95% CI)	*p*
*** Steatosis (any grade) (*N* = 80)**						
Severe psoriasis	11.9 (1.5–95.1)	0.02	12 (1.2–119.8)	0.03	10.4 (1.1–95.7)	0.04
HCV coinfection	1.8 (0.6–5.1)	0.31	1.5 (0.3–7.7)	NS		NS
Age	1.1 (1–1.1)	0.04	1 (1–1.1)	NS		NS
Time since diagnosis of HIV infection	1.1 (1–1.1)	0.04	1 (0.9–1.1)	NS		NS
Metabolic syndrome	11.9 (1.5–95.1)	0.02	8.9 (1–81)	0.05	11.9 (1.4–101)	0.02
CD4 cell count	0.99 (0.97–1.0)	0.6	1.02 (0.99–1.04)	NS	1.01 (0.99–1.03)	NS
Time of EFV exposure (years)	1.2 (1–1.4)	0.05	1.2 (1–1.4)	NS	1.2 (1.0–1.5)	0.05
**^Ç^ Significant fibrosis (*N* = 76)**						
Steatosis	2.3 (0.8–6.8)	0.12	0.9 (0.2–4)	NS		NS
Severe psoriasis	3.6 (1.3–1.)	0.02	2.2 (0.5–9.7)	NS		NS
HCV coinfection	5 (1.9–13.4)	<0.01	4.4 (1.1–18.3)	0.04	3.4 (1.1–10.6)	0.03
Age	1.1 (1–1.2)	0.01	1.1 (1–1.2)	0.09	1.1 (1–1.1)	0.09
Time since diagnosis of HIV infection	1.1 (1–1.1)	<0.01	1 (0.9–1.1)	NS		NS
CD4 cell count	0.96 (0.94–0.98)	<0.01	0.98 (0.95–0.99)	0.04	0.97 (0.95–0.99)	0.01
Time of EFV exposure (years)	1.2 (1.1–1.4)	<0.01	1.2 (1–1.3)	0.05	1.2 (1–1.3)	0.04
**^&^ Advanced fibrosis (*N* = 76)**						
Steatosis	3.5 (0.7–16.7)	0.12	2.5 (0.3–20.4)	NS		NS
Severe psoriasis	4.3 (1.4–13.4)	0.01	2.3 (0.4–12)	NS		NS
HCV coinfection	26.3 (5.4–129)	<0.01	27.4 (3.5–214)	<0.01	20.2 (4–102)	<0.01
Age	1 (1–1.2)	0.29	1.1 (0.9–1.1)	NS		NS
Time since diagnosis of HIV infection	1 (1–1.2)	<0.01	1 (0.9–1.1)	NS		NS
CD4 cell count	0.97 (0.95–0.99)	<0.01	0.98 (0.95–1.01)	0.15	0.98 (0.95–0.99)	0.046
Time of EFV exposure (years)	1.1 (1–1.3)	0.03	1.1 (0.9–1.2)	0.44		NS

* Steatosis was defined as indicative abdominal ultrasound findings, CAP > 238 dB/m, and/or TyG > 8.38. ^Ç^ Significant fibrosis measured by TE ≥ 7.1 kPa and/or FIB-4 > 1.3 for patients without HCV infection and 1.45 for patients with HCV infection. ^&^ Advanced fibrosis was measured by TE ≥ 9.6 kPa and/or FIB-4 ≥ 2.67 for patients without HCV infection and ≥3.25 for patients with HCV infection.

## Data Availability

Due to data protection regulation, data regarding the independent study are not publicly available.

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
