# Peer review of "Psoriasis and Liver Damage in HIV-Infected Patients"

_cells, 2021, doi:10.3390/cells10051099_

Round 1

Reviewer 1 Report

The manuscript by Busca-Arenzana, et al., focuses on describing the prevalence of psoriasis and liver damage in a cohort of well-controlled HIV-infected patients, and analyses risk factors associated with the development of liver complications in this population. The study shows some limitations due to its experimental design, but it provides interesting findings reporting a high prevalence of these two complications and potential associations between liver disease progression and severe psoriasis or efavirenz time exposure. As psoriasis and liver damage are important co-morbidities usually reported in HIV-infected patients, identification of the risk factors involved may be extremely relevant for their clinical management, contributing to their prevention and improving the quality of life of these patients. Some points should be addressed:

Specific comments:

  • Do the clinical histories of these patients include any other data representative of the inflammatory status of the individuals, such as cytokine plasma levels? As all the three entities under study show an inflammatory component, it would be interesting to explore its correlation with severe psoriasis and advanced fibrosis.
  • Authors describe that exposure time to the non-nucleoside reverse transcriptase inhibitor efavirenz is significantly associated with liver injury (steatosis and/or fibrosis) in HIV-infected patients with diagnosed psoriasis. Have similar effects been detected with other non-nucleoside reverse transcriptase inhibitors or is this association drug-specific?
  • The manuscript would benefit from a more-in-depth discussion of the potential pathophysiological mechanisms involved in the association between chronic HIV infection, psoriasis and NAFLD.
  • Authors should include the definitions of all the abbreviations employed in the abstract and table captions in order to make them self-explanatory.
  • Manuscript should be revised in order to correct some spelling mistakes.

Author Response

  • Do the clinical histories of these patients include any other data representative of the inflammatory status of the individuals, such as cytokine plasma levels? As all the three entities under study show an inflammatory component, it would be interesting to explore its correlation with severe psoriasis and advanced fibrosis.

C.Busca: Thank you for your comments, we agree it would be very interesting to have some subrogate marker of inflammation. Unfortunately, we have no data because they are not routinely measured in clinical practice.

  • Authors describe that exposure time to the non-nucleoside reverse transcriptase inhibitor efavirenz is significantly associated with liver injury (steatosis and/or fibrosis) in HIV-infected patients with diagnosed psoriasis. Have similar effects been detected with other non-nucleoside reverse transcriptase inhibitors or is this association drug-specific?

C.Busca: We analyzed the effects of rilpivirine on steatosis and fibrosis and we did not find association between them.

  • The manuscript would benefit from a more-in-depth discussion of the potential pathophysiological mechanisms involved in the association between chronic HIV infection, psoriasis and NAFLD.

C.Busca: We have commented on the relationship between the potential pathophysiological mechanisms involved in the association between chronic HIV infection, psoriasis and NAFLD in the introduction.

  • Authors should include the definitions of all the abbreviations employed in the abstract and table captions in order to make them self-explanatory.

C.Busca: Done. The definitions of the abbreviations are explained in the text.

  • Manuscript should be revised in order to correct some spelling mistakes.

C.Busca: Done.

Reviewer 2 Report

Abstract: far too many abbreviations, which are not defined in the abstract.

Similarly, high level use of abbreviations in the text, including some that are not defined: e.g. define FIB-4, what F2 liver means, HCV, HIS, HIS (?) etc

Table 1: the term ‘non-severe psoriasis’ is a bit unusual. ‘Mild-moderate psoriasis’ or ‘less-severe’ psoriasis would be better (note: from the patient perspective, there is relatively poor correlation between PASI and reduction in quality of life). I’m not sure the value of quoting p values when not significant – better to use NS, which makes the table less cumbersome.

The inclusion on N values makes the table confusing to read. Perhaps best to just have n values once in the heading (All: n=80 Mild-Moderate psoriasis: n=58 – Severe psoriasis: n=22)

Some spelling errors: eg. transffusion, US spelling of hemophilia,

Variability in use of ‘,’ and ‘.’ Between numbers

What was the PASI in the 10 patients with severe psoriasis who had joint involvement?

I personally find little value in statements such: To our knowledge, this is the first ….’ So what!

There is alot of data in this study, with multiple statistical testing, raising the risk of these being positive by chance.

Author Response

  • Abstract: far too many abbreviations, which are not defined in the abstract.

C.Busca: Done

  • Similarly, high level use of abbreviations in the text, including some that are not defined: e.g. define FIB-4, what F2 liver means, HCV, HIS, HIS (?) etc

C.Busca: DOne. The abbreviations are defined in the text and tables.

  • Table 1: the term ‘non-severe psoriasis’ is a bit unusual. ‘Mild-moderate psoriasis’ or ‘less-severe’ psoriasis would be better (note: from the patient perspective, there is relatively poor correlation between PASI and reduction in quality of life). I’m not sure the value of quoting p values when not significant – better to use NS, which makes the table less cumbersome.

C.Busca: Done (in the table)

  • The inclusion on N values makes the table confusing to read. Perhaps best to just have n values once in the heading (All: n=80 Mild-Moderate psoriasis: n=58 – Severe psoriasis: n=22)

C.Busca: Done in the table.

  • Some spelling errors: eg. transffusion, US spelling of hemophilia,

C.Busca: Reviewed and corrected

  • Variability in use of ‘,’ and ‘.’ Between numbers

C.Busca: Reviewed and corrected

  • What was the PASI in the 10 patients with severe psoriasis who had joint involvement?

C.Busca: PASI was > 10 in 22,2% of patients with joint involvement

  • I personally find little value in statements such: To our knowledge, this is the first ….’ So what!

C.Busca: Reviewed and corrected

  • There is a lot of data in this study, with multiple statistical testing, raising the risk of these being positive by chance.

C.Busca: We understand that multivariate analysis models reduce the risk of finding significant associations by chance. In addition, we adjusted the number of variables included in the analysis models to the events collected, maintaining a ratio of 10 to one.

Round 2

Reviewer 1 Report

I have no further suggestions

Author Response

Minor corrections:

In the Conclusions: "as during the pre-antiretroviral treatment era" should stand "as durind the period prior to the availability od combined antirretroviral treatment": done.

On page 2. "Dysregulation in T lymphocytes". Please make it clear wjat is dysregulated. Is it the number of these cells? Their function?-->"Dysregulation in T lymphocytes": mainly as a result of systemic depletion of regulatory T cells and in other situations through viral RNA replication within the CD4 + dermal dendritic cell

Please define NAFLD the first time it appears in the text and later use the abbreviation only: done

Please define ART the first time it appears in the text and later use the abbreviation only.: done

On page 3, It should stand: Fibrosis was considered significant (F≥2) if the transient elastography finding was ≥7.1 kPa and/or FIB-4 was >1.3  or >1.45 for patients without  or with HCV infection respectively; advanced liver fibrosis (F≥3) was defined as TE>9.6 kPa and/or FIB-4≥2.67 or ≥3.25 for patients without or with HCV infection respectivel: done

Please use HCV (define abbreviation the first time it appears) or hepatitis C virus throughout the whole text. It is not "hepatitis C coinfection", rather it is "hepatitis C virus coninfection": done.

On page 7, it should stand " few of the patients in our cohort": done